# Thalassemia Intermedia: Chelator or Not?

**DOI:** 10.3390/ijms231710189

**Published:** 2022-09-05

**Authors:** Yen-Chien Lee, Chi-Tai Yen, Yen-Ling Lee, Rong-Jane Chen

**Affiliations:** 1Department of Medical Oncology, Tainan Hospital, Ministry of Health and Welfare, Executive Yuan, Tainan 70043, Taiwan; 2Department of Internal Medicine, College of Medicine, National Cheng Kung University Hospital, Tainan 70403, Taiwan; 3Department of Internal Medicine, Tainan Hospital, Ministry of Health and Welfare, Executive Yuan, Tainan 70043, Taiwan; 4Department of Food Safety/Hygiene and Risk Management, College of Medicine, National Cheng Kung University, Tainan 70403, Taiwan

**Keywords:** thalassemia intermedia (TI), β-TI, HbH disease, iron chelation therapy, iron overload, non-transfusion-dependent thalassemia (NTDT), liver iron concentration (LIC), pulmonary hypertension

## Abstract

Thalassemia is the most common genetic disorder worldwide. Thalassemia intermedia (TI) is non-transfusion-dependent thalassemia (NTDT), which includes β-TI hemoglobin, E/β-thalassemia and hemoglobin H (HbH) disease. Due to the availability of iron chelation therapy, the life expectancy of thalassemia major (TM) patients is now close to that of TI patients. Iron overload is noted in TI due to the increasing iron absorption from the intestine. Questions are raised regarding the relationship between iron chelation therapy and decreased patient morbidity/mortality, as well as the starting threshold for chelation therapy. Searching all the available articles up to 12 August 2022, iron-chelation-related TI was reviewed. In addition to splenectomized patients, osteoporosis was the most common morbidity among TI cases. Most study designs related to ferritin level and morbidities were cross-sectional and most were from the same Italian study groups. Intervention studies of iron chelation therapy included a subgroup of TI that required regular transfusion. Liver iron concentration (LIC) ≥ 5 mg/g/dw measured by MRI and ferritin level > 300 ng/mL were suggested as indicators to start iron chelation therapy, and iron chelation therapy was suggested to be stopped at a ferritin level ≤ 300 ng/mL. No studies showed improved overall survival rates by iron chelation therapy. TI morbidities and mortalities cannot be explained by iron overload alone. Hypoxemia and hemolysis may play a role. Head-to-head studies comparing different treatment methods, including hydroxyurea, fetal hemoglobin-inducing agents, hypertransfusion as well as iron chelation therapy are needed for TI, hopefully separating β-TI and HbH disease. In addition, the target hemoglobin level should be determined for β-TI and HbH disease.

## 1. Introduction

Thalassemia is among the most common genetic disorders, at 1.67 percent of the population [1]. Thalassemia syndromes are characterized by reduced synthesis of globin chains, and can be phenotypically classified based on the requirements of regular blood transfusions in the first or second year of life (thalassemia major (TM)), or are largely transfusion free (thalassemia minor, thalassemia intermedia (TI) or non-transfusion-dependent thalassemia (NTDT), including β-TI [2], hemoglobin E/β-thalassemia and α-TI (HbH disease) [3]. Distinctions between these groups are based on the clinical criterion for transfusion dependency. TM patients require blood transfusions before age 2 to survive. As time passes by, TI might become transfusion dependent, usually after age 2. In a study based on 165 TI patients, 47.5% had never received blood transfusions, 25.5% transfused occasionally and 27% were regularly transfused [4]. After a longitudinal follow-up of 88 patients with TI, the mean age to start regular blood transfusions was 38 [5].

Two different mechanisms play a role in iron overload in TM and TI [6]. In TM, frequent transfusion leads finally to iron overload and cardiac toxicities, while in TI, increasing iron absorption toxicity to blood vessels, liver and hypoxemia induced heart failure [6], particularly with aging. Iron chelation therapy resulted in better overall survival in TM patients [7]. However, the role of iron chelation therapy in TI remains poorly understood, especially in those without regular transfusions. Many complications of TI have been attributed to iron overload even at the age of 5 without transfusion [8]. Pulmonary hypertension has been associated with iron overload; however, a potent inhibitor of cyclic guanosine monophosphate-specific phosphodiesterase-5 has been shown to decrease PHT in TI [8]. Iron overload might not be the single etiology for pulmonary hypertension. Iron chelation therapy has been suggested to be given to NTDT patients aged ten years or older (or fifteen years or older in deletional HbH disease) that have liver iron concentration levels of 5 mg Fe/g or over dry weight (or serum ferritin level ≥ 800 ng/mL) [9]. Iron chelation therapy will only be given by showing that decreasing iron load leads to decreased morbidities or increased overall survival rate. In addition, iron chelation therapy is not without side effects. Long-term usage of deferasirox, an oral form iron chelator, has dose-dependent renal, gastrointestinal and hepatic toxicity in addition to high costs [10,11].

β-TI and HbH disease may have similar degrees of anemia, but hemolysis not ineffective, as erythropoiesis a plays major role in HbH disease [9]. Iron overload is much more common in β-TI than HbH disease [9]. This review mainly focuses on the role of iron chelation therapy on NTDT, based on currently available evidence, related to morbidities and overall survival. The article focuses on clarifying the following questions:How can transfusion in NTDT be regularly defined?What are the complications related to iron overload in NTDT?Are iron chelators really needed in NTDT?

## 2. Difference between Transfusion-Dependent Thalassemia and Non-Transfusion-Dependent Thalassemia

NTDT shows mild to moderate anemia, having hemoglobin levels between 6 to 10 g/dL [12]. The thalassemia Clinic Research Network Registry classifies patients who received < 8 packed red blood cell (pRBC) transfusions in the past 12 months as NTDT patients [12]. Generally, TM is restricted to those patients who are dependent on transfusion for survival before the age of 2. However, later in life NTDT might become transfusion-dependent thalassemia (TDT). Blood transfusion has been proposed in β-TI even with Hb around 7–10 g/dL due to ineffective erythropoiesis of α-globin [13]. Clinical severity varies in TI cases. Patients may be asymptomatic until adult life or present symptoms between the ages of 2 and 6 years [14]. Many modifiers affect expression of TI phenotypes. One cannot predict TI phenotypes by gene expression alone. Using phenotype score to grade the severity of TI has been proposed [15] but without definite conclusions. Patients who sustain a satisfactory childhood period of Hb levels greater than 7 g/dL need to be re-evaluated at the age of 10 to 12 [14]. At this crucial age, marrow expansion and other complications are most likely reversible and with less alloimmunization of frequent transfusion [16]. Chelation therapy should be accompanied once transfusion is initiated. However, those without transfusion had lower hepcidin level and absorbed more iron, with complications later in life. In the OPTIMAL CARE study, 302 of 584 TI patients (51.7%) were regularly transfused, 143 of 584 (24.5%) occasionally transfused and 139 of 584 (23.8%) never transfused [16]. The definition of regular transfusion varies. Some define it as when a patient has received at least four blood transfusions per year [5]. Those receiving regular transfusion are treated as TM patients. They suffer from similar complications as TM patients [16]. Those who are not regularly transfused suffer from pulmonary hypertension. Pulmonary hypertension could be avoided by transfusion, iron chelation therapy and hydroxyurea treatment [16]. Another country that adapted less aggressive transfusion therapy showed that 20.4% of patients experienced pulmonary hypertension [17]. Pulmonary hypertension is best confirmed by right heart catheterization but is usually limited by real world practice and is often instead diagnosed by cardiac echo. Of those without any transfusions, iron chelation or fetal hemoglobin induction therapy, after 10 years of follow-up, cumulative multiple morbidity-free survival of those with a hemoglobulin level ≥ 10 g/dL compared with those < 10 g/dL was 100% vs. 55.3% [18], and those with a lower hemoglobin had a 6.89-fold increased risk of development of morbidity compared to that of a higher hemoglobin. Neo TDT has been proposed to patients receiving regular transfusion later in life [19]. To make it more complicated, an “itTDT (intermittently transfused TDT) has also been proposed [19].

## 3. Different Mortalities between TM and TI

After the introduction of iron chelation therapy in 1965, there was no longer an apparent difference in age of death in TM compared to TI patients [20]. In TM patients, death before 12 years of age was common before the mid-1960s, caused by anemia complicated by infection ± hypersplenism [21]. In the late 1960s, after the development of iron chelation therapy, deaths from cardiac complications in patients between 12 and 24 years of age were due to untreated or minimally treated iron overload [21]. Survival past 24 years of age was attributed to iron chelation therapy [21]. The survival of TM patients improved from poor adherence (regular transfusion and iron chelation therapy) of 15 to 25 years to good adherence, likely to at least 50 years [22]. Assessment of iron overload has been suggested after the transfusion of 10 U pRBC in patients with TDT [23]. Other studies started iron chelation therapy while patients received eight or more blood transfusions per year [24] or after a serum ferritin threshold >1000 ng/mL [23]. The main decreasing morbidities were diabetes of 46% difference followed by cardiac disease of 36% difference [25]. Other experts showed that iron chelation therapy resulted in improved survival rate and quality of life as well as reversal of hepatic and cardiac functional complications in patients with TDT, but that reversing endocrine disease was unpredictable [23]. The hazard ratio of death in TM compared to TI patients decreased from 6.8 (CI 2.6–17.5) to 2.8 (CI 0.8–9.2) after the introduction of iron chelation therapy [20]. In a study in Iran of 5491 patients, the median survival ages were 57 years old for TM and 55 years old for TI [26].

Cardiac complications due to iron overload have been reported as the cause of death in 71% of TM patients [27]. The median survival age of TM (57 years old) has improved since blood transfusion and the introduction of iron chelation survival [26], approaching the median survival of TI (55 years old) [26]. Skipping iron chelation therapy even after iron overload is brought back under control could lead to complications in TM patients [21]. Eighteen TI patients died during follow-up in Lebanon, with the causes of death being cardiac, renal and liver problems which were partially explained by iron overload [28]. However, the study did not mention whether these patients received regular transfusions or not [28]. A study that included 379 patients reported the deaths of 40 patients with TM and 13 patients with TI, and the causes of heart-related deaths were 40% vs. 15.4% in TM and TI patients, respectively [20]. Cancer was the main cause of death in five TI patients (38.2%), followed by infection in three patients (23.1%) and heart damage in two patients (15.4%) [20] (Table 1). Malignancies have been reported among NTDT patients [29]. Lacking proper control groups with staging information, the incidence changes compared with the general population remain unknown [30].

Cardiac toxicities are the main determinant of survival in TM [31] and the third in TI (rank second after cancer) [20]. Congestive heart failure, mainly left (and right) ventricular dysfunction caused by transfusion-induced iron overload, was encountered in TM patients, while pulmonary hypertension happened only in TI patients without receiving chelation transfusion [32]. Cardiac siderosis seems to be uncommon in TI, even in patients with severe iron overload [33]. Pulmonary hypertension developed in 65 of 110 TI patients (59.1%) [31]. Of all of the patients, 67 (60.9%) had not been transfused or were minimally transfused [31]. All of these patients had normal left ventricular contractility and chronic hypoxemia, and hemolysis-induced mechanisms and thromboembolism were possible causes of pulmonary hypertension [31,34]. In a study of age-matched groups of TM (n = 131) and TI (n = 74) patients, well-treated TM patients did not develop pulmonary hypertension, in contrast to TI patients [34]. In 20 never or minimally transfused patients with thalassemia intermedia (<10 units in total), no evidence of cardiac siderosis was reported [35]. Several study articles have reported that less transfused, splenectomized, hydroxyurea-naïve and iron-chelation-naïve patients developed PHT [36,37].

In an Italian study [38], including 4943 thalassemia minor and 21,063 controls, there were increasing risks of hospitalization for cirrhosis, kidney disorders, cholelithiasis and mood disorders, but no differences in death. Iron overload was suspected to be the basis of these morbidities [38] but the proportion of iron chelation therapy was unknown. In a recent study of 2033 patients with NTDT, the median age of death was 46.3 years, which might be due to shorter survival rates from the Middle East and Asia [39]. In TI, liver iron overload occurs at slower rate than in transfusion-dependent thalassemia, making it a cumulative process with advancing age. Cardiovascular disease was the leading cause of early death (median age 34.2 years), while hepatic disease or malignancies were the leading causes of death in older patients (median age 55.4, 54 years, respectively) [39]. In addition, 12.5% of patients with a median age of 10 years who were on regular transfusion died [39]. The survival rates of regularly transfused patients were better than non-regularly transfused patients [39]. Iron chelation therapy used in all regularly transfused patients and 67.7% of non-regularly transfused patients did not affect all-cause mortality nor mortality from cardiovascular disease but lowered the mortality rates from hepatic disease [39]. In a multivariate cox regression model, regular transfusion decreased mortality by around 80% in all-cause mortality and mortality from cardiovascular disease, while iron chelation therapy decreased the mortality of hepatic disease by 75% [39]. Clear guidelines on initiation of chelation therapy for TI are unclear and based on data extrapolated from TM. Regarding mortalities, TM patients suffered from cardiac morbidities, while TI patients suffered from early-age cardiovascular disease, later followed by hepatic diseases and cancers.

## 4. Different Morbidities between TM and TI

TM and TI differ in morbidities. Leg ulcers, thrombotic events, pulmonary hypertension, gallstones and extramedullary hematopoiesis were more common in patients with TI while hypogonadism, diabetes mellitus, hypothyroidism and cardiopathy were more common in patients with TM [36]. For example, in one study, thrombotic events occurred in 4% of 2190 TI patients and 0.9% in 6670 TM patients [36]. Several review articles focus on these complications and pathophysiology mechanisms [9,40,41]. NTDT could be classified according to transfusion history (Table 2). There are more studies regarding regular transfusion and morbidities of pulmonary hypertension than studies including only non- regularly transfused patients (Table 2, 33.3%, 11% vs. 1.9%). One study reported that pulmonary hypertension was also the main cause of death in cardiovascular diseases, with 36.6% of 2033 cases of NTDT, which included some of the regular transfusion population later in life [39]. Of 138 TI cases defined as having received ≤ 8 pRBC transfusions per year over a 3-year period, splenomegaly with or without splenectomy was the highest morbidity, accounting for around 50%, followed by endocrine disorder, hepatic disorders and bone disease [12]. Cardiac disorder accounted for around 5% with pulmonary hypertension accounting for around 1% [12]. In addition to splenectomized patients, osteoporosis was the most common morbidity among this patient group (Table 2).

In the human body, red blood cells contain mostly α-globin and β-globin chains which must be balanced, and the relative excess of the β- or α-chain leads to α-thalassemia or β-thalassemia, respectively [43]. Ineffective erythropoiesis is an increase in erythroid cells which fails to produce a corresponding increase in RBCs [43]. Then, iron absorption is increased but the iron is deposited in the organs rather than generating more erythrocytes [43]. The α-chain is highly unstable and unable to form soluble tetramers, the pathophysiology of which is mainly ineffective erythropoiesis. The β-chain is more stable and mature, which leads mainly to hemolysis and minimally to ineffective erythropoiesis. That is, there are no α4 tetramers owing to the high instability of the α-chain compared with HbH disease (denatured β4 tetramers): β-thalassemia is characterized by an excess of α-chain which leads to ineffective erythropoiesis, and α-thalassemia is characterized by an excess of β-chain which leads to predominately hemolytic anemia. Hyperabsorption of iron is probably not a prominent feature of hemolytic anemia with more efficient erythropoiesis, such as HbH disease [44]. Iron loading is more common in β-TI than in HbH disease [45]. Studies showed that HbH disease had lower MCV (61 (58.8–65.6) fl) and ferritin (203 (99–279) μg/L) values and higher hepcidin (2.4 (1.4–4.1) nM) compared with β-thalassemia intermedia MCV (80 (73.0–87.5) fl), ferritin (471 (298–589) μg/L) and hepcidin (0.25 (0.25–0.25) nM), respectively [45]. β-TI patients in one study had higher incidences of short stature, extramedullary erythropoiesis, pulmonary hypertension and iron overload compared to those with HbH disease (Table 3A) [46], while another study reported no difference [12] (Table 3B). No studies showed that iron chelation therapy improved overall survival rate (Table 4 and Table 5).

Do we have to wait for the complications to come or did iron chelation therapy prevent all these complications with improvement of overall survival in TI patients? It is essential to determine whether iron overload results in clinical sequelae before chelation therapy can be advised. Iron chelation therapy has been suggested to be the cornerstone of managing β-TI patients [6] and is recommended in NTDT patients older than 10 years of age who have iron overload. Is there any strong evidence to support [23]? Unfortunately, no studies showed iron chelation therapy did improve overall survival. We cannot be sure that iron chelation therapy did indeed increase survival of TI patients. 

## 5. Underlying Mechanisms of Iron Overload in TI

While blood transfusion is the main source of iron overload in regularly transfused TM patients, ineffective erythropoiesis contributes to iron overload in NTDT. Ineffective erythropoiesis leads to lower hepcidin, the only liver-producing hormone negative controlling ferroportin. Hepcidin is the master regulator of iron. It binds to ferroportin, an iron deliverer, and degrades it into lysosome. Ferroportin controls iron metabolism by expressing in enterocytes, macrophages and parenchymal hepatic cells [43]. In TI, high erythropoietic drive causes decreased hepcidin, which increases hyperabsorption of dietary iron and depletes iron storage in macrophages, resulting in “iron loading anemia”. In contrast, in a study of TM, transfusion decreased erythropoietic drive with higher hepcidin, which decreased dietary absorption with increasing iron storage in macrophages [44]. Frequent transfusion is the main reason for iron overload in thalassemia major, while it increases iron absorption in thalassemia intermedia. Special iron stains showed that stored iron is mainly found at the biliary pole of hepatocytes in TI and in hypertrophic Kupffer cells in TM [44]. Lower serum ferritin in TI reflected iron accumulation predominately in hepatocytes rather than in macrophages in TI. With similar liver iron concentrations, serum ferritin was much lower in TI than TM [44]. For example, transfusion-independent and nonchelated TI patients had lower serum ferritin levels (1316.8 ± 652.3) compared with TM patients (3723.8 ± 2568.8), but showed similar liver iron concentrations (15.0 ±7.4 vs. 15.7± 9.9, *p* = 0.095) [2]. Once TI patients were given regular blood transfusions, they suffered from the same mechanism of iron overload as TM patients.

The hepcidin level is usually lower in β-TI than in HbH disease, which might explain the reason for more clinical severity of β-TI [57]. There were several review articles regarding the regulation of hepcidin [58,59]. Lower hepcidin increased ferroportin insertion with increased gastrointestinal absorption and increased the release of iron from the reticuloendothelial system (Figure 1). Several factors control hepcidin (Figure 1). Iron is then stored in hepatocytes [44]. Ferritin levels representing iron storage in reticuloendothelial cells usually do not reflect the true iron storage amount in hepatocytes, especially in TI patients [44] (Figure 2). Measuring ferritin levels requires only one simple blood test and is much more cost effective than liver biopsy or MRI measuring of cardiac iron burden. The threshold of starting iron chelation therapy should be lower in NTDT compared with TM patients [44].

The relationships between ferritin, liver iron burden and cardiac burden were not well defined. Lack of correlation between LIC and cardiac burden has been reported [60]. Even a biopsy of the heart cannot truly reflect the iron burden in cardiac siderosis [61]. In one study, the confidence interval of ferritin and liver iron burden was large [62]. The T2 * MRI measurement has also been reported to not correlate with liver iron burden [63,64]. Liver steatosis may also interfere with ferritin measurements [65]. Cardiac siderosis, which is the main mortality of TM, plays a small role in TI [62].

## 6. Iron Chelation Therapy and Morbidities in TI

There was a total of eight studies related to ferritin level and morbidities [31,33,42,47,48,49,50,51]. Two of them were from the same source [33,48]. Most study designs regarding ferritin level and morbidities were cross-sectional (Table 4). Most studies were from groups in Italy and Lebanon [16,33,42,47,48,52,53,54,66] (Table 4, Table 5 and Table 6), such as the ORIENT study [42], THALASSA trial [52] and the OPTIMAL CARE study [16]. In a cross-sectional study of 168 TI patients without any chelation therapy, the receiver operating characteristic (ROC) curve showed the relationship between the presence and absence of morbidity: ≥7 mg Fe/g dw in vascular thrombosis; ≥6 mg Fe/g dw in pulmonary hypertension, hypothyroidism, hypogonadism and endocrine/bone disease; and ≥9 mg Fe/g dw in osteoporosis [33].

Generally, the same LIC tends to correlate with lower serum ferritin in NTDT compared with TDT [2,23]. As in morbidities, these TI groups of patients were more precisely defined as not having received regular blood transfusions (Table 4). In the ORIENT study [42], none of the TI patients who received blood transfusion, iron chelation or fetal hemoglobin induction therapy suffered from any type of morbidity, with ferritin levels of 1048.2 ± 71.2 ng/mL compared with those without, 328.7 ± 54.2 ng/mL. The mean serum ferritin level was steadily and significantly increased during the 11-year observation period from mean 513.2 ± 51.2 ng/mL (range 17–1508 ng/mL) to mean 1209 ± 103 ng/mL (range 76–2981 ng/mL) [42]. The Cox model was used, including age as a factor, and serum ferritin level was the only statistically significant variable associated with morbidity [42]. LIC cut-off was defined from 5 mg/g dw to 9 mg/g dw from MRI measurements to cause morbidities, while serum ferritin level was defined as > 300 ng/mL (Table 4). Studies reported decreasing ferritin and decreasing LIC (Table 5) but without mentioning end organ damage [52,53,54,55,56]. Studies regarding decreasing ferritin and therefore decreasing morbidities had more heterogeneous groups, including some regularly transfused patients later in life [66,67,68,69] (Table 6). Iron chelation therapy has been shown to decrease pulmonary hypertension in some studies [16,68], but did not improve heart failure, thrombosis, leg ulcers, diabetes or hypothyroidism [16] (Table 6). One study reported that blood transfusion decreased cardiac morbidities even more than iron chelation therapy [39].

The OPTIMAL CARE study [16] showed that transfusion therapy was protective for thrombosis, extramedullary hematopoiesis, pulmonary hypertension, heart failure, cholelithiasis and leg ulcers. Some morbidities were adjusted for age > 35 years old while some were not [16]. Though TI complications increased with age as well as ferritin [70], we could not tell if there were causal relationships between them. Iron chelation therapy is still given to 1 in 36 HbH and 23 out of 38 β-TI patients currently, according to clinical practice [45].

## 7. Side Effects of Iron Chelation Therapy

Sensorineural hearing loss at high frequencies with recruitment around 38%, especially with young adults, and correlation with mean and peak desferrioxamine doses administered have been reported [71]. Desferrioxamine might cause renal function to deteriorate [72]. In addition, overly vigorous chelation might lead to bone dysplasia [1].

## 8. Current Consensus and Ferritin Level

HbH disease is a type of NTDT. Even at similar levels of anemia, iron overload is more common in TI disease than in HbH disease (serum ferritin level 471 μg/L vs. 203 μg/L, respectively) [45]. Iron chelation therapy was initiated with serum ferritin levels > 1000 ng/mL [73] in several studies. There are variable ferritin level suggestions for iron chelation therapy (Table 4). It has been suggested that in patients with TI it should rarely exceed 1000 ng/mL [73,74]; however, a multicenter study on TI including 67 non-transfused or minimally transfused patients showed that they had mean peak ferritin levels of 1526 ± 578 ng/mL [31].

After the age of ten, a considerable proportion of patients might reach clinically significant levels of liver iron [42]. Chelation therapy has been suggested to be initiated when liver iron concentration reaches 5 mg Fe/g dry weight (dw) or due to ferritin levels over 800 ng/mL and stopped at 3 mg Fe/g dw or at ferritin levels below 300 ng/mL [42]. There are no studies separating the start of chelation therapy regarding ferritin levels for β-TI patients with HbH disease.

## 9. Limitations

Iron chelation therapy is not without any side effects and has been linked to a non-progressive increase in serum creatinine level [75]. Studies reporting on morbidities from iron overload in TI have increased but studies are lacking on mortalities. Ten years have been passed since the call for prospective clinical trials on TI and chelation therapy. Current practice still follows recommendations based on observational studies [16]. Many factors contribute to pulmonary hypertension; it is not explained by iron overload alone. Blood transfusion has shown to decrease pulmonary hypertension greater than iron chelation therapy. There is no head-to-head comparison between different methods for decreasing pulmonary hypertension. Some have advocated for using ferritin as a monitor while others advocate using LIC instead [33]. True evidence of target-organ iron toxicity can only be manifested through iron chelation therapy decreasing morbidity or mortality. Ten years have passed since the recommendations for the management of patients with TI were introduced, including regular assessment of liver iron concentration via biopsy or noninvasive imaging methods as well as iron chelation therapy initiation [2]. Even though all the iron-related toxicities are theoretical, we still cannot deny the truth that TI patients live shorter lives, with a median survival of 55 years old, which is even lower than the 57 years of TM patients [27].

Is TI like TM, in that if patients do not receive chelation therapy, it would lead to complications even after iron overload is brought back under control again [21]? In view of the scarcity of endocrinological and densitometric data on the β-TI population in the literature, unlike the well-studied β-TM patients, our findings warrant further studies to evaluate the impact of transfusion and iron chelation therapy on the outcomes of patients.

## 10. Summary and Future Directions

Head-to-head studies comparing different treatment methods, including hydroxyurea, fetal hemoglobin-inducing agents, hypertransfusion as well as iron chelation therapy are needed for TI, hopefully separating β-TI and HbH disease. In addition, the target hemoglobin level for β-TI and HbH should be determined.

## Figures and Tables

**Figure 1 ijms-23-10189-f001:**
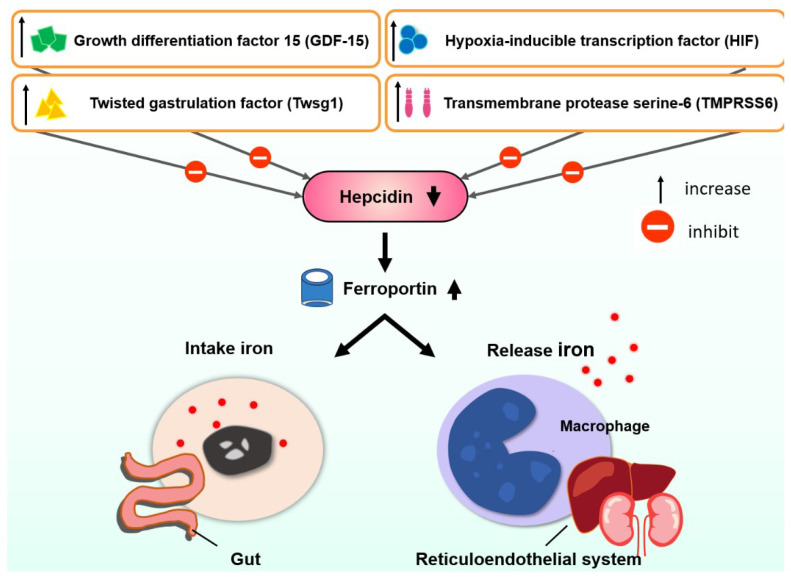
The regulation of hepcidin in thalassemia intermedia.

**Figure 2 ijms-23-10189-f002:**
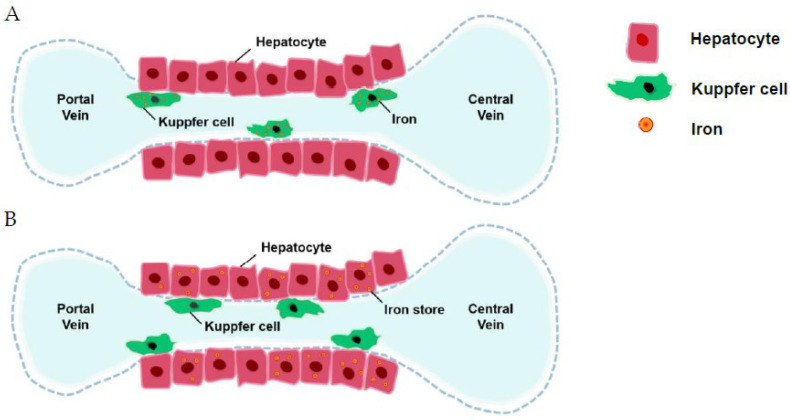
Traditionally, iron is stored in the reticuloendothelial system (**A**), but in thalassemia intermedia, iron is stored in hepatocytes (**B**).

**Table 1 ijms-23-10189-t001:** Death in thalassemia major (TM) vs. thalassemia intermedia (TI) patients from Vitrano A et al. [20].

Causes of Death %	TM, n = 40	TI, n = 13
Cancer	7.5	38.2
Heart damage	40	15.4
Infection	7.5	23.1
Multi organ failure	2.5	0
Stroke	2.5	0
Liver failure	7.5	7.7
Other complications not related to thalassemia	5	7.7
Not available	27.5	7.7

**Table 2 ijms-23-10189-t002:** Morbidities of thalassemia intermedia.

Studies, n (%)	ORIENT Study, No Iron Chelation or Fetal Hemoglobin Induction; n = 52 (β-TI) [42]	KM Musallamb, 2011, n = 168 (β-TI) [33]	OPTIMAL CARE Study, 2010, n = 584 TI [16]
Age (years), mean (SD)	24.1 ± 1.6 (range 2–56)	35.2 (12.6)	25.44 ± 13.86 (2–76)
Male, n (%)	25 (48.1)	73 (42.9)	291:293
Splenectomized, n (%)	30 (57.7)	121 (72.0)	325 (55.7)
Transfusion history, n (%)	No regularly blood transfusion		
None		44 (26.2)	139 (23.8)
Occasional		80 (47.6)	143 (24.5)
Regular		44 (26.2)	302 (51.7)
Age (years), mean (SD)	24.1 ± 1.6 (range 2–56)	35.2 (12.6)	25.44 ± 13.86 (2–76)
Male, n (%)	25 (48.1)	73 (42.9)	291:293
Splenectomized, n (%)	30 (57.7)	121 (72.0)	325 (55.7)
Morbidity, n (%)	36 (69.2)		
Osteoporosis	25 (48.1)	77 (45.8)	134 (22.9)
Extramedullary hematopoiesis	10 (19.2)	43 (25.6)	124 (21.2)
Liver disease	9 (17.3) (No B, C carrier)	54 (32.1)	57 (9.8)
Hypogonadism	4 (7.7)	28 (16.7)	101 (17.3)
DM	4 (7.7)	6 (3.6)	10 (1.7)
Thrombosis	3(5.8)	44 (26.2)	82 (14)
Pulmonary hypertension	1 (1.9)	56 (33.3)	64 (11)
Hypoparathyroidism	1 (1.9)		
Leg ulcers		41 (24.4)	46 (7.9)
Hypothyroidism	5 (9.6)	30 (17.9)	33 (5.7)
Heart failure		9 (5.4)	25 (4.3)
cholelithiasis			100 (17.1)

**Table 3 ijms-23-10189-t003:** (**A**) Differences between HbH and β-thalassemia [46]. (**B**) Differences between HbH and β-thalassemia [37].

(**A**)				
**Complications %**	**HbH, n = 72**	**β-TI, n = 80**	**<18 Years (n = 104)**	**>18 Years (n = 48)**
Short stature	22.2	42.5		
Growth retardation	34.7	50		
Osteoporosis	18	30	24	25
Extramedullary hematopoiesis	5.6	17.5	12.5	10.4
Heart failure	0	13.8	6.7	8.3
Pulmonary hypertension	1.4	5	2	3.3
Gallstones	1.4	2.5	0	6.3
Cholecystectomy	0	7.5	1	10.4
Deep vein thrombosis	0	2.5	0	4.1
Leg ulcers	1.4	3.8	2	4.1
Iron overload	11.1	45	30.8	25
(**B**)				
**Complications %**	**HbH, n = 84**	**β-TI, n = 39**	**Age < 10 Years, n = 94**	**Age ≥ 10 Years, n = 44**
Spleen (splenomegaly ± splenectomy)	55	51	49	66
Bone (osteopenia, osteoporosis, ± bone deformity)	17	36	12	41
Endocrine	32	26	21	52
Hepatic	25	15	16	36
Gallbladder	7	5	1	16
Cardiac	6	8	6	7
Pulmonary hypertension	1	3	1	2
Infections	25	15	21	18
Thrombosis	1	5	1	7

**Table 4 ijms-23-10189-t004:** Ferritin level and related morbidities.

Study	Study Subject	N, Study Design	Morbidity Ferritin Level (Patient Number)/Patient Number (%)		Note
Alberto Roghi et al., 2010 [47]	TI, regularly transfused (2–4 times/year) Infrequently (few) Transfusion naïve Iron chelation therapy ≥ 1 years Italy	49 age ≥ 18 years (range 23–64) Iron chelation Yes (n = 34), No (n = 15) Cross-sectional	No evidence of cardiac iron overload Lack correlation between cardiac T2 and LIC or serum ferritin Statistically significant positive but poor linearity between serum ferritin and LIC Multivariate analysis: LIC positive with ALT levels		No defined morbidity
The ORIENT study [42]	β-TI, age ≥ 2 years Hb 7–9 g/dL without regular BT, exclude iron chelation or fetal hemoglobin induction therapy before or throughout the observation period; no death or lost follow-up 5 centers in Middle East and Italy	52 patients 11-year Retrospective cohort	≥800 (n = 27) 300–800 (n = 17) ≤ 300(n = 8) Liver disease (non-B, C) 27(100%) 9 (52.9%) 0 Any type of morbidity Endocrine 15 (55.6%) 4 (23.5%) 0 Extramedullary 23 (85.2%) 6 (35.3%) 0 hemostasis, thrombosis, PHN adjusted for age, sex, splenectomy, mean Hb, ferritin ↑ 1 ng/mL: hazard ratio at least 1 morbidity 1.002, *p* = 0.002, multiple morbidity 1.011, *p* = 0.005		Mean annual % Hb ↓ 0.3%; mean ferritin ↑ 9%
KM Musallam 2011, 2013 [33,48]	β-TI, age ≥ 2 years with Hb 7 to 9 g/dL without regular transfusion at diagnosis No iron chelation or fetal hemoglobin-inducing agents 2 centers in Italy Transfusion history: 1. Every 1–3 m, regularly transfused 2. Occasionally transfused (severe anemia secondary to infection, surgery, or pregnancy) 3. Non-transfused	168 patients Cross-sectional	LIC cut-off (mg/Fe/g) dw(MRI):		adjusted for age, gender, transfusion status, splenectomy, etc.
			osteoporosis	≥9	
			pulmonary hypertension	≥6	
			thrombosis	≥7	
			hypothyroidism	≥6	
			hypogonadism	≥6	
			vascular	≥7	
			endocrine/bone	≥6	
			LIC ≥ 5 mg/g dw vs. < 5 mg/g dw (MRI):		
			osteoporosis	(58.2 vs. 28.6%)	
			pulmonary hypertension	(43.9 vs. 18.6%)	
			thrombosis	(34.7 vs. 14.3%)	
			hypothyroidism	(24.5 vs. 8.6%)	
			hypogonadism	(23.5 vs. 7.1%)	
F E Chen et al., 2000 [49]	HbH Chinese patient review chart Jan 1998 to Dec 1999	114 patients Cross-sectional	Ferritin increased with age (*p* < 0.001) not related to transfusion history 6 patients liver biopsy for ↑ ferritin or abnormal liver function (all HBV(−), HCV(−), herbal(−), long-term iron therapy(−)), fibrosis in 5 and 2 out of 5 cirrhosis		9/80 HBsAg (+); 0/80 anti-HCV Ab(+)
Luke K L Chan et al., 2021 [50]	HbH Chinese patients ≥ 18 years 1 HCV carrier 1 alcoholic Transfusion history 35, iron chelation 9	80 patients Cross-sectional	Advanced liver fibrosis (transient elastography) Univariable, clinically significant age ≥ 65 years, moderate-to-severe liver overload (LIC ≥ 7 mg/g dry weight, MRI), serum ferritin 800 ≥ µg/L multivariable regression with clinical significance age ≥ 65 years, moderate-to-severe liver overload		
Kunrada Inthawong, 2015 [51]	NTDT < 3 RBC/year in the last 5 years, age 10–50 years Exclude secondary pulmonary hypertension	76 patients Retrospective cohort	12 iron chelation Previous splenectomy, higher cumulative RBC transfusions (≥10 RBC) Nucleated RBC, a high non-transferrin-bound iron, but not ferritin or LIC associated with pulmonary hypertension (9.2%)		
Aessopos A et al., 2001 [31]	60.9% not transfused or minimally transfused; rest start transfusion after age > 5 year Group A: no or rare transfusion Group B: occasional transfusion	110 β-TI Retrospective cohort for 2 years	Start Ferritin > 1000 ng/mL Stop < 1000 ng/mL Multivariate regression analysis: pulmonary hypertension not related to serum ferritin level, peak ferritin level, transfusion, etc.		

**Table 5 ijms-23-10189-t005:** Decreased ferritin, but do not mention morbidities.

Study	Study Subject	N, Study Design	Morbidity Ferritin Level (Patient Number)/Patient Number (%)	Note:
THALASSA trial, 2012 [52] THALASSA trial, 1 year extension 2013 [53] Ali T Taher et al., 2014 [54]; Part of THALASSA trial	NTDT ≥ 10 years and LIC ≥ 5 mg Fe/g dw (MRI) and Ferritin > 300 ng/mL in 2 consecutive values ≥ 14 days; 1 year result No transfusion within 6 months or chelation therapy within 1 month Exclude HBV, HCV carrier NTDT age ≥ 10 years LIC ≥ 5 mg Fe/g dw and Ferritin > 300 μg/mL Deferasirox-naïve, no other chelation therapy within 1 month of study entry Deferasirox vs. placebo 1 year and then 1 year extension placebo switched to deferasirox	166 patients Phase 2, prospective, randomized, double-blind, placebo-controlled trial 130 patients 130 patients	Start LIC ≥ 5 mg Fe/g dw and Ferritin > 300 ng/mL Suspended Ferritin < 100 ng/mL or LIC < 3 mg Fe/g dw ↓ LIC significantly, ↓ serum ferritin level significantly ↓ LIC significantly, ↓ serum ferritin level significantly Trend decreasing in mean ALT Trend increase serum creatinine and decrease creatinine clearance; increase urinary protein/creatinine ratio initially and then stable Ferritin 800 ≥ μg/L predict LIC ≥ 5 mg Interruption (ferritin <300 μg/L) Dose escalation (ferritin > 2000 μg/L)	
Pootrakul P et al., 2003 [55]	Deferiprone 1 year in Thailand Transfused while Hb < 5 g/dL	7 β-TI/HbE, 2 β-TI	Significant fall in liver iron, serum ferritin, red cell membrane iron and serum non-transferrin bound iron	3 transfused 6 not transfused
Vassilis Ladis et al., 2010 [56]	Ferritin 500 ≥ µg/L and LIC ≥ 2 mg Fe/g dw (MRI) NTDT, some <20 pRBC Exclude renal, liver function impairment or a life expectancy <2 years	11 TI patients Deferasirox discontinue ferritin < 250 µg/L and LIC < 2 mg Fe/g dw	1 patient excluded for 6 months of iron chelation therapy 1 patient 12 months LIC decrease, *p* = 0.044 LVEF no change	

**Table 6 ijms-23-10189-t006:** Experiment studies: decrease ferritin, decrease morbidities.

Study	Population	Study Design		Complications	
Chan JC, 2006 [67]	HbH, ferritin > 900 μg/L Control, HbH, ferritin < 900 μg/L; No chelation except one > 6 m ago	Age-matched cohort study DFO for 18 months	17 study cases with 16 control (1 case excluded due to intolerance)	Ferritin < 397 μg/L in two F/U or after 18 months, stop DFO 1. MRI for liver: an improvement in T2-signal intensity ratio 12/16 patients 2. Diastolic dysfunction (E/A ratio) no change throughout the period	None transfused
The OPTIMAL CARE study, 2010 [16]	TI, Hb> 7 g/dL at age 2 6 centers (Lebanon, Italy, Egypt) Transfusion history: 1. Every 1–3 m, regularly transfused 2. Occasionally transfused (severe anemia secondary to infection, surgery, or pregnancy) 3. Non-transfused	Retrospective cohort	584 TI, transfusion None (139) Occasional (143) Regular (302)	Morbidity *p* value Ferritin ≥ 1000 μg/L iron chelation EMH 0.85 NA PHT NA 0.53 * HF NA 0.45 Thrombosis 1.86* 0.97 Cholelithiasis NA 0.30 * Abnormal liver function 1.74 * NA Leg ulcers 1.29 0.68 DM NA 0.4, Hypothyroidism NA 0.49 Osteoporosis 1.6 0.4 * Hypogonadism 2.63 * 2.51 *	Iron chelation (+) therapy ≥ 1 year Some morbidities adjusted for age > 35 years old
Musallam KM, 2012 [66]	Transfusion-independent β-TI, exclude HCV Patients with at least two elastographies during the study period (Lebanon)	Retrospective cohort	42 patients, F/U 4 years, median age 38 years	Transient elastography values for fibrosis 2 patients improve fibrosis stage Significant change in serum ferritin, elastography value	33.3% iron chelation throughout the study period
Krittapoom Akrawinthawong, 2011 [68]	β-TI/HbE age 18–50 years, ferritin > 1000 ng/mL Transfused while Hb< 6 g/dL Exclude kidney, liver, heart disease history, BT > 3500 mL/year and ferritin > 10,000 ng/mL DFP for 1 year	Prospective cohort	30 patients unable to use deferoxamine 1 anaphylaxis shock	↓ Ferritin (*p* = 0.005) and pulmonary hypertension (*p* = 0.021) ↓ Oxidative stress markers (*p* < 0.01)	
Ersi Voskaridou, 2009 [69]	≤20 RBC unites in their lifetime Liver or cardiac iron overload (serum ferritin ≥ 1000 μg/L, liver MRI T2 <25 ms or cardiac T2 < 28 ms) Exclude pregnant, hepatic failure (transaminase > 500 U/l) or renal failure (eGFR < 60 mL/min) or LVEF< 50%	11 TI deferasirox for 12 months (one patient Hb 12.3 g/dL) (TI, not separate α- and β- thalassemia) Single-arm prospective	Liver MRI T2 * improved, GOT, GPT decreased Serum ferritin reduced Mean cardiac T2 and LVEF not change significantly		

* clinical significant; ↓, decrease; EMH, extramedullary hematopoiesis; PHT, pulmonary hypertension; HF, heart failure; DM, diabetes mellitus; DFO, deferoxamine.

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
