# Peer review of "Thalassemia Intermedia: Chelator or Not?"

_ijms, 2022, doi:10.3390/ijms231710189_

Round 1
Reviewer 1 Report
In this manuscript, the authors intend to provide a review of the current knowledge about thalassemia intermedia and iron chelation therapy. The manuscript is generally well written, with a good inclusion of available literature on the subject, citing many of the relevant papers in the field.
The authors point out a medical need regarding the lack of a consolidated and validated strategy to chelate non-transfusion dependent thalassemia patients.
They ask if NTDT patients really benefit from iron chelation therapy and say that iron chelation therapy is not without side effects. I think it’s important to remember that there’s also the risk to chelate too late, when irreversible complications have already occurred.
Some relevant information, in my view, is partially missing especially with regard to liver disease.
In thalassemia intermedia, liver iron overload occurs at slower rate than in transfusion dependent thalassemia, making it a cumulative process with advancing age. I think you should underline in the text how time is an important factor in understanding why morbidity is mainly evident in older adults. In the context of liver disease, I think you should mention the increased risk of hepatic failure resulting from hepatic fibrosis and cirrhosis as the result of damage caused by the profibrogenic effect of iron and the reversibility of liver damage (fibrosis). I think you should consider in the context of cancers the incidence of hepatocellular carcinoma in NTDT (in the study, reference n.16, TI have 2 cases of HCC out of 5 cases of cancer). Mortality for liver disease accounts for 10% in TI.
In the contest of PTH I think you should include papers where PTH is confirmed on right heart catheterization to provide more accurate data on prevalence.
Minor considerations:
Abstract line 20 please replace IT with IT
Pag2 line 54 please specify for acronym PTH
Pag2 line 64 reference 10: please can you add reference regarding side effects and costs of DFX in hemoglobinopathies?
Pag3 line 101 the number of references is 53(?), please renumber
Pag4 line 144 cardiac toxicities in the study are the third in TI
Tables 4,5,6 are very difficult to read
Some English language revision is also required.
Author Response
Reviewer 1:
In this manuscript, the authors intend to provide a review of the current knowledge about thalassemia intermedia and iron chelation therapy. The manuscript is generally well written, with a good inclusion of available literature on the subject, citing many of the relevant papers in the field.
The authors point out a medical need regarding the lack of a consolidated and validated strategy to chelate non-transfusion dependent thalassemia patients.
They ask if NTDT patients really benefit from iron chelation therapy and say that iron chelation therapy is not without side effects. I think it’s important to remember that there’s also the risk to chelate too late, when irreversible complications have already occurred.
Thank you reviewer for the helpful comment. We do notice there is a risk to chelate too late in TM patients and had mention it in page 3 of 16, line 132 “ Skipping iron chelation therapy, even after iron overload is brought back under control could lead to complications in TM patients.” And had also mention it in limitation section page 13 of 16 line 349 “Do TI like TM, if not receiving chelation therapy, would lead to complications even after iorn load is brought back under control again. “
Some relevant information, in my view, is partially missing especially with regard to liver disease.
In thalassemia intermedia, liver iron overload occurs at slower rate than in transfusion dependent thalassemia, making it a cumulative process with advancing age. I think you should underline in the text how time is an important factor in understanding why morbidity is mainly evident in older adults. In the context of liver disease, I think you should mention the increased risk of hepatic failure resulting from hepatic fibrosis and cirrhosis as the result of damage caused by the profibrogenic effect of iron and the reversibility of liver damage (fibrosis).
Thank you the reviewer for the helpful comment. We did mention about liver iron overload occurs at slower rate and liver iron overload occurs with advancing age. Page 4 of 16, line167 “Cardiovascular disease was the leading cause of early death (median age 34.2 years) while hepatic disease or malignancies were for the older (median age 55.4, 54 years respectively) [34]. 12.5% with median of 10 years patients were on regular transfusion [34]. Survival of regularly transfused patients was better than non-regularly transfused patients [34]. Iron chelation therapy used in all regularly-transfused patients and 67.7% of non-regularly-transfused patients didn’t affect all-causes mortality nor mortality from cardiovascular disease, but improved in mortality from hepatic disease [34]. “ We have added reviewer helpful comment and sentence in our text “In thalassemia intermedia, liver iron overload occurs at slower rate than in transfusion dependent thalassemia, making it a cumulative process with advancing age.” Insert before this paragraph.
I think you should consider in the context of cancers the incidence of hepatocellular carcinoma in NTDT (in the study, reference n.16, TI have 2 cases of HCC out of 5 cases of cancer). Mortality for liver disease accounts for 10% in TI.
Thank you the reviewer for the helpful comment. We did agree hepatocellular carcinoma might be a cause of death in TI. However, lacking proper control, we are not sure if the incidence really increased compared with general population owing to few cases reported. We did mention this in our paragraph. Page 3 of 16, line 143, “Cancer was the main cause of death in TI 5 patients (38.2%) following by infection 3 patients (23.1%) and heart damage 2 patients (15.4%) [16] (Table 1). Malignancies have been reported among NTDT [25]. Lacking proper control groups with staging information, the incidence changes compared with general population remained unknown [21].”
In the contest of PTH I think you should include papers where PTH is confirmed on right heart catheterization to provide more accurate data on prevalence.
Thank you the reviewer for the helpful comment. Unfortunately, we couldn’t find any paper diagnosis of PTH all by confirmed by catheterization. Almost all the studies using cardiac echo for diagnosis, the references we included with catheterization proof of PTH, only performed in 6 (heart failure) of 110 patients reference [26]. We think it’s important to prove all the PTH by gold standard diagnosis tool, but limited by the real world practice.
Minor considerations:
Abstract line 20 please replace IT with IT
We have corrected it.
Pag2 line 54 please specify for acronym PTH
We have corrected it.
Pag2 line 64 reference 10: please can you add reference regarding side effects and costs of DFX in hemoglobinopathies?
We have added one more references from reference 10 to 10 to 11.
Pag3 line 101 the number of references is 53(?), please renumber
We have renumbered it. Thank you.
Pag4 line 144 cardiac toxicities in the study are the third in TI
We cannot find this sentences.
Tables 4,5,6 are very difficult to read
We are sorry about the difficult to read. We have make it smaller and more compact and see if they become more readable.
We thank you for the reviewer taking time and much effort to review this article.
Some English language revision is also required.
We have had English speaker to revision the article and make it more readable.

Reviewer 2 Report
In the manuscript entitled "Thalassemia intermedia, to chelator or not?" Yen-Chien Lee et al. aim to gather the current knowledge regarding the use of chelating agents in thalassemia intermedia. In addition, the authors provide information for the differences between transfusion dependent and non-dependent patients, and thalassemia intermedia and HbH disease. For this purpose, the authors cite relevant bibliography and try to comment on the usefulness (if any) of chelation therapy. This Reviewer has some comments, as follow:
1. My main concern and general comment is about the structure of the Manuscript. The authors make a lot of comparisons between different sub-groups (e.g., chelation or not, transfusion dependency or not, HbH vs bTI). This information may be relevant to the scope of the article, but makes it really difficult to follow. I would advise reconstructing the Manuscript according to the comparisons needed to the authors in order to reach their main conclusions regarding iron chelation therapy in each case.
2. Abstract section: It would be preferable if the questions included in the Abstract were replaced with sentences pointing out the gaps in the current knowledge.
3. Introduction section: Please provide the definition of PHT the first time you mention it.
4. Introduction section: This Reviewer does not understand the placement of the following sentence to the end of this section “Hopefully, with enough evidence separated for β-TI and HbH diseases”. Please rephrase.
5. Paragraph 2 is entitled “Definition of TM, TI and thalassemia minor”. This title seems to not fit the content of the specific paragraph. Please rephrase it in the reorganized manuscript to reflect the information that follows.
6. Paragraph 3: At the end of this paragraph the authors raise some questions regarding iron chelation therapy. It would be preferable to replace them with a concluding remark and if possible provide plausible directions regarding this hot issue.
7. Paragraph 4: After Table 2 the authors provide information for human red blood cells in the thalassemia context. The specific part does not fit in this paragraph, but rather in paragraph 5 where the underlying mechanisms of iron overload in thalassemia are stated.
8. Paragraph 4: The authors state that “No studies showed iron chelation therapy did improve overall survival” (lines 215-216). However, in the exactly next sentence there is this question: “Do we have to wait for the complications to come or did iron chelation therapy prevent all these complications with improvement of overall survival in TI patients?” (lines 221-222). This Reviewer cannot understand if iron chelation therapy improved overall survival or not, since the above statements are contradicting. Please clarify this discrepancy.
9. Paragraph 5: Please provide the definition of LIC the first time you use it.
1. Limitations Paragraph: The authors state that “Even though all the iron related toxicity were theoretically, we still cannot deny the truth that TI lived shorter with a median survival of 55-year-old, even lower than 57 year-old of TM”. Nonetheless, in paragraph 3 it was written that: “The median survival of TM (57-year-old) had improved since blood transfusion and the introduced of iron chelation survival approaching approximately to the median survival of TI (55-year-old)”. This Reviewer believes that these sentences are a bit contradicting. Please rephrase.
Author Response
Reviewer 2:
In the manuscript entitled "Thalassemia intermedia, to chelator or not?" Yen-Chien Lee et al. aim to gather the current knowledge regarding the use of chelating agents in thalassemia intermedia. In addition, the authors provide information for the differences between transfusion dependent and non-dependent patients, and thalassemia intermedia and HbH disease. For this purpose, the authors cite relevant bibliography and try to comment on the usefulness (if any) of chelation therapy. This Reviewer has some comments, as follow:
- My main concern and general comment is about the structure of the Manuscript. The authors make a lot of comparisons between different sub-groups (e.g., chelation or not, transfusion dependency or not, HbH vs bTI). This information may be relevant to the scope of the article, but makes it really difficult to follow. I would advise reconstructing the Manuscript according to the comparisons needed to the authors in order to reach their main conclusions regarding iron chelation therapy in each case.
We thankful for the reviewer comment. But could the reviewer give us a clear or an example guideline for how to construct. So we can improve our manuscript according to the suggestions. Thank you
- Abstract section: It would be preferable if the questions included in the Abstract were replaced with sentences pointing out the gaps in the current knowledge.
These are 4 key questions that the authors tried to clarify from the article. Could the reviewer give us a good suggestion how to rephrase them? We have consulted English speaker and still couldn’t think a better idea.
- Introduction section: Please provide the definition of PHT the first time you mention it.
We thankful for the review comment. However, PHT definition varied from studies to studies. We couldn’t give a definition of PHT while other studies didn’t follow.
- Introduction section: This Reviewer does not understand the placement of the following sentence to the end of this section “Hopefully, with enough evidence separated for β-TI and HbH diseases”. Please rephrase.
We thankful for the reviewer comment and had deleted this confusing sentences.
- Paragraph 2 is entitled “Definition of TM, TI and thalassemia minor”. This title seems to not fit the content of the specific paragraph. Please rephrase it in the reorganized manuscript to reflect the information that follows.
We thankful for the reviewer comment and couldn’t think of a good title to make it clearer. Could the reviewer give us a suggestion that we can make it a better article. Thank you.
- Paragraph 3: At the end of this paragraph the authors raise some questions regarding iron chelation therapy. It would be preferable to replace them with a concluding remark and if possible provide plausible directions regarding this hot issue.
We have deleted the raise questions and replace it with “Regarding mortalities, TM suffered from cardiac morbidities while TI suffered from early age of cardiovascular disease following late by hepatic diseases and cancers. “
- Paragraph 4: After Table 2 the authors provide information for human red blood cells in the thalassemia context. The specific part does not fit in this paragraph, but rather in paragraph 5 where the underlying mechanisms of iron overload in thalassemia are stated.
This table 2 tried to specified morbidities of TI and didn’t try to explain “Underlying mechanisms of iron overload in TI, title of paragraph 5. “ Paragraph 5 is illustrated by figure 1 and 2.
- Paragraph 4: The authors state that “No studies showed iron chelation therapy did improve overall survival” (lines 215-216). However, in the exactly next sentence there is this question: “Do we have to wait for the complications to come or did iron chelation therapy prevent all these complications with improvement of overall survival in TI patients?” (lines 221-222). This Reviewer cannot understand if iron chelation therapy improved overall survival or not, since the above statements are contradicting. Please clarify this discrepancy.
Thank you the reviewer comment. We have tried to find all the reference published to included one study to prove iron chelation therapy did improve overall survival, but unfortunately found none. No one studies published till now prove that iron chelation therapy did improve overall survival as solid as e.g. cancer related research usually used overall survival as primary end point. We could only find decreasing morbidities but couldn’t be sure that by decreasing morbidities do improve survival. We have rephrase this to make the paragraph clearer by the following sentences. “Unfortunately, no studies showed iron chelation therapy did improve overall survival. We couldn’t sure that by iron chelation therapy did increase survival of TI patients.”
- Paragraph 5: Please provide the definition of LIC the first time you use it.
LIC means liver iron concentration. LIC definition varies. For example, some defined as calculated from liver T2* image (TE 0.99-16.50 ms) according to the formula [(1/T2*/1,000)]x0.0254+0.202] (PMID 20016898). So authors cannot make a definite definition of LIC while other studies have their definition.
- Limitations Paragraph: The authors state that “Even though all the iron related toxicity were theoretically, we still cannot deny the truth that TI lived shorter with a median survival of 55-year-old, even lower than 57 year-old of TM”. Nonetheless, in paragraph 3 it was written that: “The median survival of TM (57-year-old) had improved since blood transfusion and the introduced of iron chelation survival approaching approximately to the median survival of TI (55-year-old)”. This Reviewer believes that these sentences are a bit contradicting. Please rephrase.
We have mention in the paragraph 3, “In the late 1960s for the development of iron chelation therapy, causes death from cardiac complications between 12 and 24 years of age are due to untreated or minimally treated iron overload [17]. Survival past 24 years of age was attributed to iron-chelation therapy [17]. The survival of TM improved from poor adherence (regular transfusion and iron chelation therapy), 15 to 25 years longer to good adherence, probably at least 50 years [18]. “ In a more recent study, it is surprised that TM median survival is 57-year old longer than median survival of TI (55-year-old). So we try to figure out whether iron chelation therapy is needed in TI to prolong survival. “ The survival ages were very closed and that’s why we had written this article to see if iron chelation therapy really need to improved survival rates of TI.

Round 2
Reviewer 1 Report
Dear Authors,
Thank you to taking my suggestions.
I still have a concern regarding PH. I understand your point of view but the information you provide could be confusing. Please consider Guidelines for the Management of Non-Transfusion Dependent Thalassaemia (NTDT) (2nd Edition – 2017) where you can find all references for definition and epidemiology of PH and write in text you point of view (We think it’s important to prove all the PTH by gold standard diagnosis tool, but limited by the real world practice)
Pag4 line 144 "Cardiac toxicities are the main determinant of survival in TM [26] and 2nd to TI (rank second after cancer): cardiac toxicities in the study you have cited are third cause in TI and not second after infections
Author Response
Dear Authors,
Thank you to taking my suggestions.
I still have a concern regarding PH. I understand your point of view but the information you provide could be confusing. Please consider Guidelines for the Management of Non-Transfusion Dependent Thalassaemia (NTDT) (2nd Edition – 2017) where you can find all references for definition and epidemiology of PH and write in text you point of view (We think it’s important to prove all the PTH by gold standard diagnosis tool, but limited by the real world practice)
Thank you for the reviewer comment. We have added “Pulmonary hypertension is best confirmed by right heart catheterization, but usually limited by real world practice and often diagnosis by cardiac echo. “ in page 3 of 15, line 101.
Pag4 line 144 "Cardiac toxicities are the main determinant of survival in TM [26] and 2nd to TI (rank second after cancer): cardiac toxicities in the study you have cited are third cause in TI and not second after infections
Thank you the reviewer comment. We have added the reference citation as below and changed the sentence according to the evidence. “Cardiac toxicities are the main determinant of survival in TM [26] and 3rd to TI (rank second after cancer) [20]”
Reviewer 2 Report
The authors have addressed a significant part of this Reviewer’s comments and they have generally improved their manuscript. I only have a few suggestions as follow:
Regarding the reorganization of the information, since the authors have multiple comparisons in their manuscript, I would recommend adding subtitles to help the reader focus on the specific information. For example, when transfusion- dependent versus independent patients are compared: “Differences between transfusion-dependent and independent patients”. Similar subtitles could be added for comparisons between chelation therapy and no therapy or HbH and bTI subjects.
In the abstract section the questions can be avoided by replacing them with a phrase like: “Questions are raised regarding the relation between iron overload or iron chelation therapy ad patient morbidity/mortality as well as the starting threshold for chelation therapy”.
In paragraph 2, the information is more of a general description rather than a definition. In addition, the paragraph does not contain information regarding thalassemia minor. Please rephrase the title by deleting thalassemia minor and replacing definition with description.
Author Response
Reviewer 2:
The authors have addressed a significant part of this Reviewer’s comments and they have generally improved their manuscript. I only have a few suggestions as follow:
Regarding the reorganization of the information, since the authors have multiple comparisons in their manuscript, I would recommend adding subtitles to help the reader focus on the specific information. For example, when transfusion- dependent versus independent patients are compared: “Differences between transfusion-dependent and independent patients”. Similar subtitles could be added for comparisons between chelation therapy and no therapy or HbH and bTI subjects.
Thank you for the reviewer comment. We understand the point. However, some TI become transfusion dependent. We compared TM with TI instead of transfusion-dependent versus independent. They are not mutually exclusive.
In the abstract section the questions can be avoided by replacing them with a phrase like: “Questions are raised regarding the relation between iron overload or iron chelation therapy ad patient morbidity/mortality as well as the starting threshold for chelation therapy”.
We thankful for the reviewer suggestion and had changed it according to the reviewer suggestion in the abstract section “Questions are raised regarding the relation between iron chelation therapy with decreased patient morbidity/mortality as well as the starting threshold for chelation therapy. “ We also deleted all the question sentence in the abstract section. Thank you reviewer for the rephrases.
In paragraph 2, the information is more of a general description rather than a definition. In addition, the paragraph does not contain information regarding thalassemia minor. Please rephrase the title by deleting thalassemia minor and replacing definition with description.
We have changed the title to “Different between transfusion dependent thalassemia and non-transfusion-dependent thalassemia” as reviewer suggested.